# Distribution and Phylogeny of Erythrocytic Necrosis Virus (ENV) in Salmon Suggests Marine Origin

**DOI:** 10.3390/v11040358

**Published:** 2019-04-18

**Authors:** Veronica A. Pagowski, Gideon J. Mordecai, Kristina M. Miller, Angela D. Schulze, Karia H. Kaukinen, Tobi J. Ming, Shaorong Li, Amy K. Teffer, Amy Tabata, Curtis A. Suttle

**Affiliations:** 1Department of Earth, Ocean and Atmospheric Sciences, University of British Columbia, 2207 Main Mall, Vancouver, BC V6T 1Z4, Canada; v.pagowski@alumni.ubc.ca; 2Pacific Biological Station, Fisheries and Oceans Canada, 3190 Hammond Bay Rd, Nanaimo, BC V9T 6N7, Canada; Angela.Schulze@dfo-mpo.gc.ca (A.D.S.); Karia.Kaukinen@dfo-mpo.gc.ca (K.H.K.); Tobi.Ming@dfo-mpo.gc.ca (T.J.M.); Shaorong.Li@dfo-mpo.gc.ca (S.L.); Amy.Tabata@dfo-mpo.gc.ca (A.T.); 3Biology Department, University of Victoria, 3800 Finnerty Rd, Victoria, BC V8P 5C2, Canada; akteffer@gmail.com; 4Department of Microbiology and Immunology, University of British Columbia, 2207 Main Mall, Vancouver, BC V6T 1Z4, Canada; 5Department of Botany, University of British Columbia, 2207 Main Mall, Vancouver, BC V6T 1Z4, Canada; 6Institute for the Oceans and Fisheries, University of British Columbia, 2207 Main Mall, Vancouver, BC V6T 1Z4, Canada

**Keywords:** erythrocytic necrosis virus (ENV), viral erythrocytic necrosis (VEN), Pacific salmon, Pacific herring, British Columbia

## Abstract

Viral erythrocytic necrosis (VEN) affects over 20 species of marine and anadromous fishes in the North Atlantic and North Pacific Oceans. However, the distribution and strain variation of its viral causative agent, erythrocytic necrosis virus (ENV), has not been well characterized within Pacific salmon. Here, metatranscriptomic sequencing of Chinook salmon revealed that ENV infecting salmon was closely related to ENV from Pacific herring, with inferred amino-acid sequences from Chinook salmon being 99% identical to those reported for herring. Sequence analysis also revealed 89 protein-encoding sequences attributed to ENV, greatly expanding the amount of genetic information available for this virus. High-throughput PCR of over 19,000 fish showed that ENV is widely distributed in the NE Pacific Ocean and was detected in 12 of 16 tested species, including in 27% of herring, 38% of anchovy, 17% of pollock, and 13% of sand lance. Despite frequent detection in marine fish, ENV prevalence was significantly lower in fish from freshwater (0.03%), as assessed with a generalized linear mixed effects model (*p* = 5.5 × 10^−8^). Thus, marine fish are likely a reservoir for the virus. High genetic similarity between ENV obtained from salmon and herring also suggests that transmission between these hosts is likely.

## 1. Introduction

Viral erythrocytic necrosis (VEN) is a disease associated with severe blood abnormalities in infected fish which has caused mass mortality in Pacific herring (*Clupea pallasii*) [1]. The disease is traditionally diagnosed by microscopic examination of stained blood smears for the presence of inclusion bodies within the cytoplasm of infected erythrocytes. Electron microscopy revealed that infected erythrocytes contained icosahedral virions, which were named erythrocytic necrosis virus (ENV) [2]. Although first described more than half a century ago, the virus is poorly characterized as attempts to propagate it in fish cell lines have been unsuccessful [3].

Based on the small amount of available ENV genomic sequence, the virus has been assigned to a new putative genus within the *Iridoviridae* (a family of double-stranded DNA viruses), comprising other erythrocytic viruses from ectothermic hosts [2,4]. Further genomic sequencing of ENV may provide a reference for studying genetic variation geographically, and among fish species. Currently, the only verified ENV sequences available in GenBank encode ATPase, the major capsid protein (MCP), DNA-dependent DNA polymerase, and DNA-dependent RNA polymerase.

In the NE Pacific Ocean, VEN has been described in Pacific herring [1] and in the marine phase of pink (*Oncorhynchus gorbuscha*), chum (*Oncorhynchus keta*), coho (*Oncorhynchus kisutch*), steelhead (*Oncorhynchus mykiss*), and Chinook (*Oncorhynchus tshawytscha*) salmon [5,6]. High geographic variability in VEN prevalence and disease susceptibility of chum, coho, sockeye, and Chinook salmon, as well as Pacific herring, suggests that ENV could help explain high year-to-year variability in the population dynamics of these keystone species throughout coastal regions of the NE Pacific Ocean [1,7,8]. In the laboratory, VEN has been induced after confining healthy salmon with diseased fish. Among salmonids, disease transmission has been demonstrated in pink and chum salmon [5,8,9]. Chinook, coho, and sockeye salmon appear to be more resistant to infection in challenge studies, as assessed by electron microscopy [5,9], despite natural epizootics in Chinook and coho salmon. To date, an explicit relationship between disease manifestation and viral load has not been described, and ENV isolation has not been accomplished in previous studies reporting disease transmission among species. There is substantial evidence that viruses in different genera within the family *Iridoviridae* cause different disease manifestation and severity [10,11,12,13,14,15]; thus, phylogenetic placement and genetic characterization of ENV may offer insight into factors contributing to disease outbreaks.

Considering the high prevalence of ENV in herring and salmon in the NE Pacific Ocean [1,16,17,18,19,20], there is a need to further characterize the virus in at-risk fish populations. Substantial herring stock declines in British Columbia since the 1970s, which have mainly been attributed to overfishing, are of concern as herring are a keystone species that underpin the coastal food web [21,22]. Similarly, Pacific salmon provide an important biological, economic, and cultural resource in British Columbia, and recent stock declines are likely to have large impacts on wildlife and fisheries in the region. The Pacific Salmon Commission, for example, reported a 60% reduction in Salish Sea Chinook abundance from 1984 to 2010 [23]. Furthermore, the Committee on the Status of Endangered Wildlife in Canada reports that half of the British Columbia Chinook salmon populations are endangered [24].

In the current study, we applied metatranscriptomic sequencing to examine the phylogenetic relationship of ENV in Chinook salmon and investigated the epidemiology and host tropism of ENV using high-throughput PCR. Our results show that ENV is widely prevalent across numerous fish species in the NE Pacific Ocean and that ENV in salmon and herring has high genetic similarity, suggesting that the virus may circulate among these species.

## 2. Materials and Methods

### 2.1. Fish Sampling

In total, 19,652 fish comprising 16 species were sampled from freshwater and marine environments as previously described [17,25]. Briefly, 3228 freshwater samples were collected at hatcheries, through beach seining, or by smolt traps for wild salmon. Marine samples were obtained mostly by purse and beach seines, and by trawl. Typically, mixed tissue samples were dissected from fish using sterile procedures in the field [26] and frozen in RNAlater before nucleic acid extraction, although some fish were flash frozen in the field and dissected in the laboratory. Both procedures have been routinely used for examining individual fish for the presence of viral nucleic acids [16,18,26,27,28]. Samples were collected over the course of an 11-year period from 2007–2018 in a region spanning Alaska to Northern Washington (Appendix A) as part of a large pathogen-screening effort conducted by Fisheries and Oceans Canada.

### 2.2. Data Collection

The occurrence and abundance of ENV in fish was determined using the Fluidigm BioMark Platform at the Department of Fisheries and Oceans Canada [26]. Briefly, the platform provides an estimate of viral load based on copy number, as assessed by RT-qPCR. Copy numbers are calculated based on serial dilutions of artificial construct DNA standards. The calculated limit of detection (LOD) was applied to identify fish with amplifications above the 95% detection threshold [26]. ENV assays on this platform show 100% inclusivity (detection of all known strain variants of the targeted microbe) and 97.9% exclusivity (no detection of untargeted microbe species). Primers were originally obtained from a 100 bp ATPase-like protein partial gene sequence. Additional details on primer sequences as well as assay specificity and reliability are available in Miller et al. [26]. ENV monitoring was conducted alongside assays for 46 other infective agents on combined RNA and DNA extractions. In this study, ENV prevalence is reported as the proportion of fish with ENV detections, both with and without the LOD criteria applied. When not mentioned explicitly, prevalence values are reported with LOD criteria applied. Statistical analyses were done only on samples with LOD criteria applied.

### 2.3. Metatranscriptomic Sequencing and Bioinformatics

In order to isolate putative ENV sequences, several samples with high-load detections of this virus, as assessed using the Fluidigm BioMark, were selected for transcriptomic sequencing. In total, three aquaculture and one wild Chinook salmon were used to predict ENV proteins. Three additional fish, including an Atlantic salmon, a Chinook salmon, and a herring sample underwent an enrichment step for these predicted ENV proteins before bioinformatic analysis.

All fish except the wild Chinook salmon were sequenced on Illumina Next-Generation Sequencing (NGS) platforms using different RNA-seq protocols, depending on original targets and loads. To avoid DNA contamination from the host reducing the sequencing depth of the target virus, RNA sequencing was used to obtain transcriptomic sequences of the DNA virus. Each of these libraries was single tissue, either heart or spleen. Ribosomal RNA was removed from total RNA using the RiboMinus Invitrogen Eukaryote kit for RNA (Life Technologies, Carlsbad, CA, USA). The RNA-Seq library was prepared using the NEBNext Ultra RNA Library prep kit (New England BioLabs, Ipswich, MA, USA) with an average fragment size of 250-bp and was paired-end sequenced with 100-bp reads on the Illumina HiSeq analyzer (Illumina, San Diego, CA, USA).

For the wild Chinook sample, the ENV contigs were obtained using a similar RNAseq approach. This library was created using pooled tissue (gill, liver, heart, kidney and brain) and prepared with the ScriptSeq Complete Epidemiology NGS library kit (Illumina, San Diego, CA, USA). Briefly, ribosomal RNA was removed from Total RNA using the Epicentre ScriptSeq Complete Gold Kit (Epidemiology) (Illumina, San Diego, CA, USA) according to the manufacturer’s instructions. The ScriptSeq Index reverse primers were added to the cDNA during the final amplification step which involved 14 cycles. Finally, a paired-end 125 bp sequencing run was performed on the Illumina HiSeq System.

Finally, in order to enhance our sensitivity (i.e., NGS read depth and coverage) for ENV in the medium load (~1300 copies) farmed Atlantic salmon sample, and the Chinook and herring samples (~42,670 and 214,500 copies, respectively) we employed SureSelect^XT^ enrichment technology (Agilent, Santa Clara, CA, USA). A custom set of RNA target enrichment probes (120 bp in length and staggered along the exome of interest) were designed for ENV as well as many other salmonid viruses assessed on our infectious agent monitoring platform. These sequences (497.266-kbp in total length) and subsequent bait oligonucleotides included all of the suspected ENV contigs previously assembled from high load samples. Baits which failed the SureSelect QA/QC parameters and/or significantly matched salmonid genes via BLAST searches were removed, leaving the final set of enrichment probes at 20,497. The mixed tissue (gill, atrium, ventricle, liver, pyloric caeca, spleen, head kidney, posterior kidney) RNA library, was prepared using the SureSelect^XT^ low input (NGS) target workflow (Agilent, Santa Clara, CA, USA) with a SureSelect^XT^ RNA Direct/XTHS modified protocol. Approximately 200 ng of total RNA was lyophilized and fed into the SureSelect Strand-Specific RNA library Prep kit (Agilent, Santa Clara, CA, USA) according to the manufacturer’s instructions. After the 2nd strand cDNA synthesis and end repair steps, the library prep was moved to the SureSelect^XT^ low input reagent kit, starting with the end repair and A tailing step. Molecular barcoded adaptors were added using ligation and then amplified for 14 cycles according to manufacturer’s instructions to create a pre-capture RNAseq library. Samples were quantified with the Qubit dsDNA HS kit (Invitrogen, Carlsbad, CA, USA), qualified with the DNA12000 chips run on the Agilent 2100 Bioanalyzer (Agilent, Santa Clara, CA, USA), and pooled into a batch of 12 prior to hybridizing 1500 ng to the bait library. Hybridizations were incubated at 65 ℃, captured on streptavidin beads (Beckman Coulter, Brea, CA, USA) and washed at 70 ℃ according to manufacturer’s instructions before a post-capture amplification of 14 cycles. These final libraries were quantified with the Qubit dsDNA HS kit (Invitrogen, Carlsbad, CA, USA) and qualified with the DNA HS chips run on the Agilent 2100 Bioanalyzer (Agilent, Santa Clara, CA, USA). Finally, a paired-end 101 bp v2 300 kit sequencing run was performed on the Illumina Miseq (Illumina, San Diego, CA, USA), which included a 5% phiX spike in.

After adapter removal, Illumina MiSeq sequencing produced between 53.3 and 59.9 M reads for each Chinook salmon used to predict ENV amino-acid sequences (quality score >28). These reads were processed as detailed below. Adapters were removed using Trimmomatic [29], and the trimmed reads aligned with genome sequence from Atlantic Salmon [30] using the Burrows-Wheeler Aligner [31]. Unmapped sequences were extracted from the dataset using Samtools [32] and assembled into contiguous sequences (contigs) using SPAdes [33]. The translated contigs were queried against the non-redundant (NR) database in GenBank using DIAMOND [34]. Contigs with top hits to members of the family *Iridoviridae* were extracted in Microsoft Excel and the Qiime script *filter_fasta.py* [35] was adapted to retain these contigs as fasta files. GeneMark [36] was used to make protein predictions for putative ENV nucleotide sequences. Predicted proteins were subject to a BLAST search against the NR database [37], the lymphocystis disease virus genome, and a set of 47 conserved genes within Nucleo-Cytoplasmic Large DNA viruses (NCLDVs) [38]. Additionally, contigs of 500 bp in length or greater were extracted and translated into all six frames using Geneious version 9.1.8 [39]. A single, most likely translation frame was selected from the BLAST results, and those with BLAST results mapping to ENV were used to create phylogenies for ATPase, DNA-dependent DNA polymerase, DNA-dependent RNA polymerase, and the MCP. For each of these proteins, phylogenetic trees were mapped using available sequences from closely related iridoviruses using ClustalW for alignments and PhyML with Le Gascuel substitution model for tree creation [40,41]. For each phylogenetic tree, *Spodoptera frugiperda ascovirus 1a* was used as an outgroup. Assembled contigs have been submitted to GenBank under the accession numbers MK638669–MK638757. To increase confidence that sequences we attributed to ENV belong to this virus, we aligned sequences obtained from the farmed Atlantic salmon (632,935 reads), Chinook salmon (658,241 reads), and herring (800,139 reads) which underwent an enrichment step for ENV viral content to our putative ENV protein-encoding sequences using the using the Burrows-Wheeler Aligner [31].

### 2.4. Spatial Epidemiology Analysis

All statistical analyses and plots were performed in R version 3.5.2 [42]; scripts are available in Appendix A. For statistical analyses, ENV detection was categorized as positive or negative with LOD criteria applied and viral loads were quantified based on estimated viral copy numbers in host fish, following Miller et al. [26]. For map plotting only, ENV load was quantified by subdividing ENV copy number data into six categorical bins based on value with no LOD criteria (0, 0–5, 5–10, 10–100, 100–10,000, >10,000 viral copies). As sample sizes for other fish were small, statistical analyses were only conducted on herring and salmon. However, ENV prevalence and load was investigated for all sampled fish species. Heat maps were produced with an inverse-distance weighting function, using R packages “ggmap” and “gstat” [43,44], while plots were constructed with “ggplot2” [45]. Differences in load among species, age class, and habitat type were assessed using Kruskal-Wallis and post-hoc pairwise Dunn tests with Benjamini-Hochberg adjusted *p* values for multiple comparisons. Differences in ENV prevalence among categorical variables were assessed using Chi-squared tests of independence. A post-hoc Fisher’s exact test and Chi-squared tests with Bonferroni correction were conducted for pairwise comparisons to determine ENV prevalence differences among species and years, respectively, with “rcompanion” [46]. Spearman correlation was conducted to examine correlations among monthly prevalence between species. To account for residual variation in the data, a generalized linear mixed-effects model with Laplace approximation was implemented to examine differences in ENV prevalence between fresh and saltwater, with catch region, species, age class, season, population (hatchery or wild), and year considered as random effects with the R package “mlmRev” [47,48]. A similar model was used to assess differences in prevalence among age classes (smolt or adult) with habitat type used as an additional random effect instead of age class.

## 3. Results

### 3.1. Genetic Characterization

Metatranscriptomic sequencing of Chinook salmon with high ENV loads revealed high sequence identity between ENV sequences found in Chinook salmon and viral sequences from GenBank associated with Pacific herring. The available ENV sequences from Pacific herring (ATPase, DNA-dependent DNA polymerase, MCP, and DNA-dependent RNA polymerase) showed over 99% nucleic acid identity to ENV sequences from Chinook salmon in our study (Table 1). Sequences reported in Table 1 originate from heart tissue of one of the aquaculture Chinook salmon samples and from the wild Chinook salmon mixed-tissue specimen, both with a high load ENV detection (CT values of 10.7 and 13.4, respectively). The aquaculture Chinook salmon specimen had jaundice and several co-infections including *Paranucleospora theridion*, Piscine reovirus, and *Renibacterium salmoninarum,* as determined by the RT-qPCR assay. Phylogenies based on these genes (Figure 1) from ENV and its relatives (Table 2), showed that viral sequences from herring and salmon form a well-supported clade. A metatranscriptomic approach on a DNA virus only reveals virally expressed transcripts; thus, a full ENV genome could not be assembled. Within the metatranscriptomic sequences, there were 32 putative ENV proteins which consistently mapped to proteins of similar function from viruses in the family *Iridoviridae* (Appendix A), as well as BLAST hits to 21 of the 47 core proteins in NCLDVs [38] (Appendix A). Putative ENV transcripts typically had between 20% and 60% nucleotide identity with other iridoviruses. These values are consistent with nucleotide identities reported between existing ENV sequences and other fish iridoviruses. In total 117 contigs with BLAST hits to iridoviruses were isolated from transcriptomic sequencing. Putative ENV sequences ranged in length from 152 to 4475 bp (File S3). From these, we predicted 89 protein-encoding sequences longer than 200-bp (Accession numbers MK638669-MK638757). When putative ENV protein-encoding sequences were aligned to reads obtained from the Atlantic salmon, Chinook salmon, and Pacific herring enriched for these contigs, we detected 35 of 117 putative ENV sequences. Results from this analysis are summarized in Appendix A.

### 3.2. Spatial Epidemiology

ENV is widely distributed in the NE Pacific Ocean and was detected in 12 of 16 tested species (Figure 2) throughout the sampling region. High viral loads were common within the Strait of Georgia, along the west coast of Vancouver Island, and in straits and channels throughout coastal northern British Columbia and southern coastal Alaska (Figure 3). Over 19,000 fish were tested for ENV, from 16 different species collected from marine and fresh waters spanning from Washington to Alaska. ENV prevalence was highest in anchovy and herring, occurring in over 27% of all sampled fish in these species and in over 37% of smolts. In herring, the proportion of fish in which ENV was detected was significantly higher than in any salmon species tested. We also report significantly lower ENV prevalence and load within salmon smolt (p_prev_ = 1.2 × 10^−17^, p_load_ = 1.8 × 10^−4^) and greater ENV prevalence among herring smolt (*p* = 4.5 × 10^−6^), when compared to respective adult prevalence. Among fish with detections of ENV, viral load was highest in herring and Chinook salmon and lowest in Atlantic salmon. ENV load also appeared low in chum and pink salmon; however, these differences were not statistically significant (Appendix A, Appendix A). Significant differences in ENV load were found between herring and coho, Chinook, sockeye, and Atlantic salmon (*p* values are given in Appendix A).

Despite high ENV prevalence and load in coastal regions, ENV was rarely detected, and percent prevalence was statistically lower in all species sampled in freshwater (*p* < 2.2 × 10^−16^) (Figure 4). Furthermore, after applying a generalized linear mixed-effects model with Laplace approximation to account for residual variation in the data, categorical classification of habitat type as freshwater or saltwater still had a significant effect on ENV prevalence (*p* = 5.50 × 10^−8^). Given that 98% of the freshwater detections were below the LOD, these all represent very low-load detections, and are unlikely biologically relevant or may represent “false” positives. As such, all statistical tests are reported for samples with LOD criteria applied only.

### 3.3. Seasonal and Yearly Variation

In Atlantic salmon, ENV prevalence showed seasonal variation, with the lowest prevalence occurring in August, for both smolt and adult fish (Appendix A). A similar seasonal trend was observed in smolt sockeye and Chinook salmon. A Spearman correlation coefficient of 0.87 was found between monthly prevalence of ENV in sockeye and Atlantic salmon (*p* = 0.004). Despite very low ENV prevalence in Atlantic salmon (2%), farmed coho and Chinook salmon showed higher ENV prevalence (29% and 48%, respectively) compared to their wild counterparts (3% and 7%, respectively) (Appendix A). This difference was significant in Chinook salmon (*p* = 2.08 × 10^−50^). Among salmon, changes in ENV prevalence from year to year appear to occur synchronously (Appendix A) and ENV prevalence also varies significantly by year (*p* < 2.2 × 10^−16^, Appendix A). A 63% decrease from the average prevalence of 5% is observed after 2013.

## 4. Discussion

The current study expands on sequence available for ENV and demonstrates that the virus is widespread in salmon and herring and is often present at high load in these fish. The low genetic variation among viruses infecting salmon and herring has implications for potential host range and the taxonomic classification of these viruses.

Nucleotide variation of ENV from herring and Chinook salmon was low, with ENV sequences from Pacific herring [2,4] having >99% nucleotide identity with the sequence obtained from Chinook salmon in this study, indicating that both viruses belong within the same genus, and likely the same species. High sequence similarity between herring and Chinook salmon could be suggestive of viral spillover between hosts. Moreover, the phylogenetic analysis clearly places ENV within the *Iridoviridae*, but distinct from other viruses within the family.

After enrichment for putative ENV sequences in the herring, Chinook salmon, and Atlantic salmon samples, we found 35 of 117 predicted protein-encoding sequences, suggesting that not all ENV transcripts are present in every infected fish. This may depend on expression levels and infection stage. Indeed, variable gene expression during different stages of infection occurs commonly among iridoviruses [49,50,51,52]. Moreover, our estimates of prevalence are conservative, as the assay is sensitive to sequence variation, so related strains of ENV could be missed.

ENV was widely distributed geographically and among fish species in marine waters of western North America. Despite high ENV prevalence and load in coastal marine environments, viral load and prevalence were consistently low in freshwater environments. Of the 3622 fish analyzed from freshwater, ENV was only detected in one Chinook salmon specimen after LOD criteria were applied. Interestingly, there were few ENV detections in fish from the open waters north of Vancouver Island (Figure 3); whereas ENV was common in coastal areas, i.e., channels, inlets, and straits along coastal British Columbia and southern Alaska (Figure 3). We hypothesize that interspecies transmission may be more likely in these areas, where there are higher densities of salmon and other fish. Greater ENV prevalence in aquaculture Pacific salmon than in wild counterparts (Appendix A) supports the idea that transmission of the virus may be more common in coastal and high-density environments. There is some support for this hypothesis in the literature as previous studies have shown that chum salmon may contract VEN via waterborne exposure [8] and that increased stock density is a predictor of disease progression in fish infected with other iridoviruses [53,54]. We also report significantly lower ENV prevalence among salmon smolts and in adult herring when compared to respective adult and smolt age classes for these species, suggesting that susceptibility to ENV varies by host species and age class. ENV prevalence may not, however, directly correlate with disease manifestation mediated by the virus, as species with low ENV load and prevalence (<0.5%) in this study, including chum and pink salmon, are more susceptible to VEN than other salmon species [55,56]. Thus, salmon species in which ENV prevalence and load were the greatest (Chinook and coho salmon) may be able to sustain higher viral loads and display fewer clinical symptoms.

We hypothesize that salmon likely contract the virus from marine reservoirs, given the low detection of ENV in freshwater salmon, high sequence similarity between ENV in salmon and herring, and high viral prevalence in several species of marine fish including herring, anchovy, pollock, and sand lance. Furthermore, the prevalence of ENV in Pacific salmon is much higher than in Atlantic salmon, suggesting that farmed Atlantic salmon are not a significant reservoir for ENV transmission, despite high-density rearing in this species. Previous research has shown that overall infectious agent diversity and burden increases when sockeye salmon enter the ocean [16], suggesting that ENV could contribute to increased infection stress experienced by out-migrating salmon smolts.

Salmon migration, which varies among species and populations, may help explain yearly and seasonal variation in ENV prevalence. Peaks in ENV prevalence occur during spring and late fall for Chinook and sockeye salmon, with significant drops seen during peak salmon river runs in July and August. A similar pattern is observed in Atlantic salmon, which are stationary throughout the year and therefore may provide a useful sentinel to study seasonal ENV dynamics in wild salmon. Indeed, there is a significant correlation in monthly ENV prevalence between Atlantic and sockeye salmon. However, variation in migration patterns among species and stocks complicates the analysis of seasonal ENV prevalence changes. Migration routes and timing among Chinook salmon stocks, for example, is highly variable [57,58]. Furthermore, several sockeye salmon stocks were sampled during different parts of the year. Increases in ENV prevalence in winter may also arise as a result of herring migrations to coastal regions during this time [59]. Overall, monthly ENV prevalence dynamics in salmon were similar to those observed previously in herring [1], further substantiating the hypothesis that interactions with herring promote infection dynamics in wild salmon.

Previously, little was known about the epidemiology of ENV in salmon and marine fish, as most studies focused on ENV in herring. Hershberger et al. detected ENV in up to 67% of herring, with similar seasonal variation to that which we observed in salmon, with the greatest proportion of fish testing positive for ENV in summer months [1]. They also reported that ENV epizootics can arise and dissipate spontaneously in geographically isolated regions along the North Pacific coastline. Additionally, Teffer et al. investigated ENV prevalence in returning Chinook salmon and detected ENV in 16% of tagged males and 25% of females in the Chilliwack River [20]. Together, our research and these studies indicate that the virus is widely distributed on the west coast of British Columbia and Alaska and that salmon are likely infected once they enter the ocean, with herring or other marine fish likely acting as a reservoir for ENV.

### Implications

Detection of ENV has not been conclusively linked to disease onset and further studies are required to characterize this relationship. VEN is a poorly characterized disease in Chinook and sockeye salmon, yet it is relatively common in wild at-risk populations of these species. In contrast, ENV was relatively rare in pink and chum salmon, even though these species are more susceptible to VEN than Chinook and sockeye salmon in challenge studies [5]. However, relatively few pink (*n* = 222) and chum (*n* = 191) salmon were sampled in our study. There were significant differences in viral loads among species, with lowest mean ENV loads occurring in Atlantic salmon. Low viral loads and prevalence could indicate higher virulence, which may lower the chance of transmission and detection compared to persistently infected fish [60]. Alternatively, species with large ranges in ENV copy number, such as herring and Chinook salmon, could carry a persistent infection which becomes virulent at higher loads. These species may transmit the virus to susceptible species such as pink and chum salmon. Future challenge studies which further characterize Chinook salmon infection may elucidate whether infection dynamics appear similar to those of herring, which have the most similar distribution of viral load.

ENV prevalence was lower in salmon smolts and adult herring. Similarly, Hershberger et al. reported more frequent VEN epizootics in juvenile herring, compared to adults [1]. Presumably, lower viral prevalence in smolt salmon arises because fewer smolt have been exposed to marine waters, where we propose the virus originates. Previous studies reported that osmoregulatory stress, such as transitions between saline and freshwater environments, could be implicated in herring mortality in fish infected with VEN [15,61]. If viral infection does, indeed, impact osmoregulatory capacity and adaptation, the relatively high prevalence and load of ENV detected in salmon soon after ocean entry could diminish their ability to properly acclimate to changes in salinity in their environment.

Numerous studies [10,11,12,13,14] have reported greater disease severity caused by iridoviruses that are closely related to ENV when the temperature increases. It has been suggested that below 20 °C, iridoviruses may remain dormant in teleost hosts [14]. Other members of *Iridoviridae* that infect fish and show high infection mortality typically occur in warmer climates, such as Southeast Asia and Australia. Similarly, VEN progression is most severe during the summer in Pacific salmon [5]. Changes in temperature were not investigated in this study, but seasonal and yearly variation in ENV prevalence suggests that environmental variables, such as temperature, may be important. A significant drop in ENV prevalence following 2013 coincides with a shift to a positive Pacific Decadal Oscillation Index and warming temperatures in the study region [62].

It is possible that disease progression intensifies at warmer temperatures, such that fewer fish harboring the virus survive. This interpretation is consistent with a decrease in the overall prevalence of salmon infectious agents in the region from 2012 to 2013 reported by Nekouei et al. [16]. Alternatively, fish infected with the virus may be weakened or more susceptible to other diseases at suboptimal temperatures. In the context of climate change, this is an interesting avenue of future research, and directly relevant to salmon populations, as evidence suggests that increasing coastal and oceanic temperatures can have significant and detrimental effects on salmon migration and spawning [63,64]. If VEN has a temperature-dependent onset similar to other diseases caused by iridoviruses, ENV-mediated mortality could further stress at-risk populations of salmon and herring in the NE Pacific Ocean.

## 5. Conclusions

This research demonstrates that ENV is highly prevalent in the NE Pacific Ocean. Low ENV prevalence in freshwater, high prevalence in marine fish, and seasonal variability corresponding to marine migrations of salmon and herring suggest that ENV originates from the marine environment. High prevalence in several marine fish species suggests that the virus is endemic and that these species are reservoirs of the virus. Moreover, the similarity between ENV sequences from Chinook salmon and those from Pacific herring indicates that transmission between these species is possible. Finally, we present 89 new protein-encoding sequences attributed to ENV in this study.

## Figures and Tables

**Figure 1 viruses-11-00358-f001:**
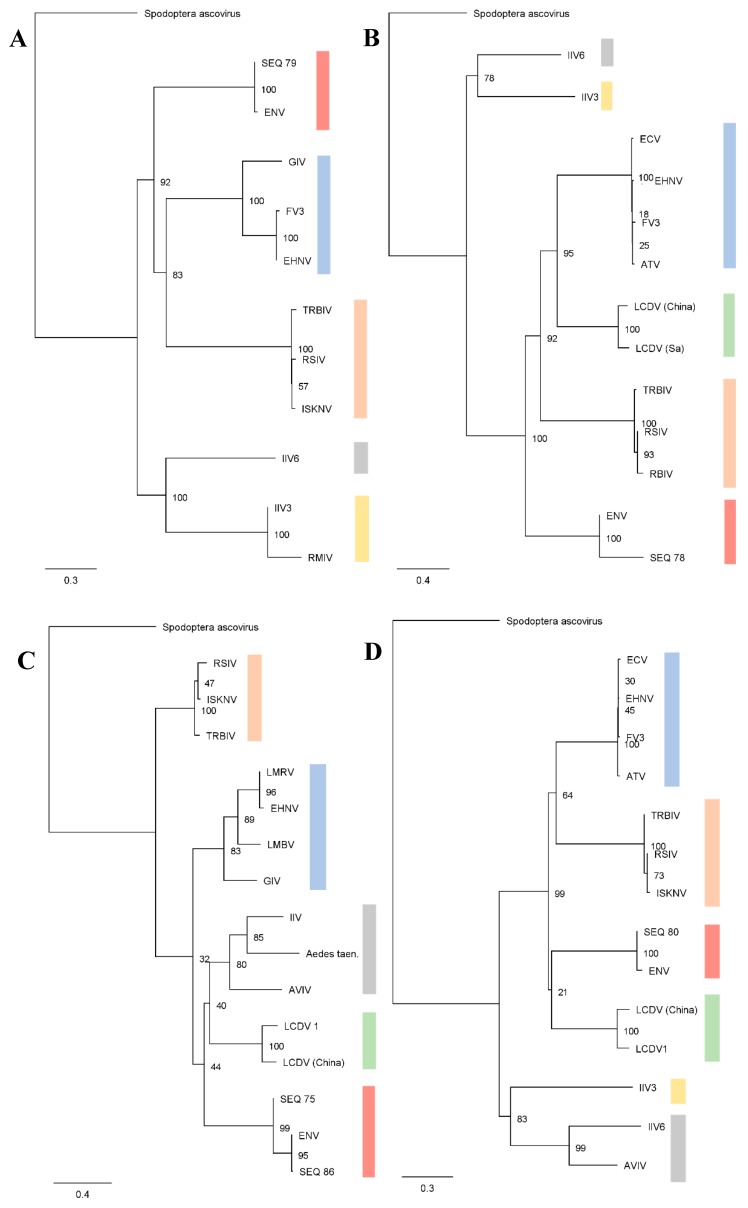
Substitution-rate optimized maximum likelihood phylogenetic trees of putative ENV DNA-dependent DNA polymerase (**A**), DNA-dependent RNA polymerase (**B**), major capsid protein (**C**), and ATPase (**D**) sequences (SEQ#) mapped with related iridoviruses. Alignments were based on nucleotide sequences for the major capsid protein and amino acid sequences for all other trees. Branch labels indicate bootstrap support values for 100 re-samplings and the scale bar indicates substitution rate. GenBank reference sequences and contig IDs are listed below (Table 2), with colors indicating genera groups. Putative ENV sequence lengths are listed in Table 1.

**Figure 2 viruses-11-00358-f002:**
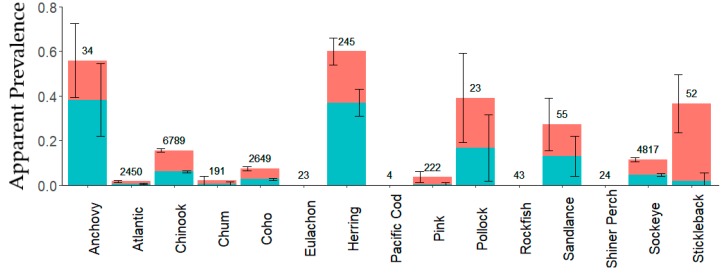
Apparent ENV prevalence in mixed tissue smolt samples by species. Error bars indicate 95% confidence intervals. Blue bars indicate proportions with limit of detection (LOD) criteria applied and coral bars indicate proportions without LOD criteria applied. Values indicate sample sizes for each species.

**Figure 3 viruses-11-00358-f003:**
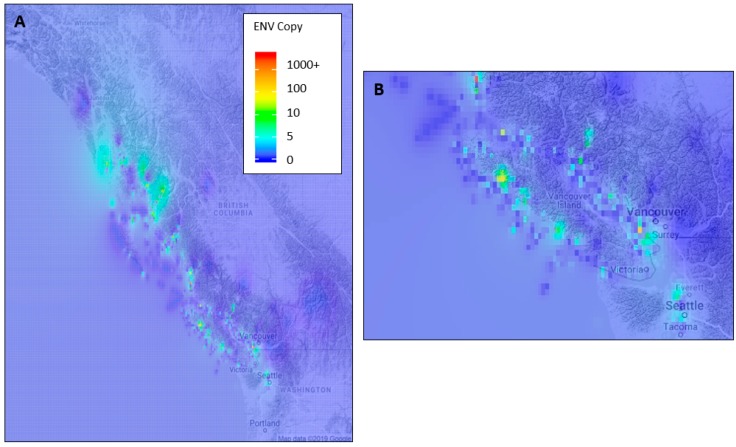
(**A**) Full sampling area heat map of the mean calculated ENV copy number, based on interpolated values. Copy number values are binned into 6 numerical categories (see color legend), (**B**) shows Vancouver Island inset map. Adults and smolts of all species are shown and LOD criteria are not applied.

**Figure 4 viruses-11-00358-f004:**
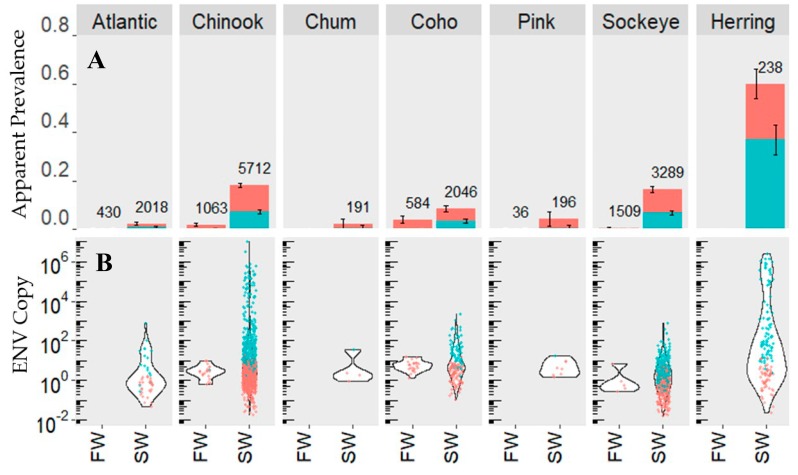
Saline (SW) and freshwater (FW) ENV prevalence (**A**) and load (**B**) among salmon species and herring. Samples with LOD criteria applied are in blue and samples without LOD criteria applied are in coral. Printed values indicate sample sizes and error bars indicate 95% confidence intervals. Only smolts are shown.

**Table 1 viruses-11-00358-t001:** **Basic Local Alignment Search Tool** (BLAST) summary of erythrocytic necrosis virus (ENV) sequences from Chinook salmon compared with available herring derived ENV sequences on GenBank. The origin of each sequence is indicated as aquaculture (Aq.) or wild Chinook salmon. Sequences used in phylogenies are in bold.

Reference Sequence	Sequence ID	% Identity (Nucleotide)	% Identity (Amino Acid)	Sequence Alignment Length (AA)
ATPase KJ730210.1 (partial)	**80 (Aq.)**	99.9	100	849
	46 (Wild)	99.6	98.7	74
	57 (Wild)	99.5	98.5	67
	54 (Wild)	100	100	77
DNA-dependent DNA polymerase KJ756347.1 (partial)	73 (Wild)	99.9	98.3	59
	**79 (Aq.)**	99.8	99.7	897
	19 (Wild)	99.5	99.5	213
MCP KT211480.1 (partial, Puget Sound)	**86 (Aq.)**	99.6	100	1436
	**75 (Wild)**	100	100	175
DNA-dependent RNA polymerase KJ756346.1 (partial)	**78 (Aq.)**	99.9	96.8	972

**Table 2 viruses-11-00358-t002:** GenBank reference sequences and viral species used to create phylogenies, with colors indicating genera groups corresponding to phylogenetic trees in Figure 1.

Abbreviation	Accession Numbers				Taxon
	DNA-Dependent DNA Polymerase	DNA-Dependent RNA Polymerase	MCP	ATPase	
NA	AAC54632.1	YP_762407.1	NC_008361.1	NC_008361.1	Spodoptera ascovirus
ENV	AIQ77732.1	AIQ77731.1	KT211480.1	AIN76233.1	Erythrocytic necrosis
**Megalocytivirus**					
RSIV	BAA28669.1	BAK14252.1	AB109371.1	BAK14298.1	Red Sea Bream iridovirus
ISKNV	CAZ73994.1		AF370008.1	NP_612331.1	Infectious spleen and kidney necrosis virus
TRBIV	ADE34365.1	ADE34378.1	AY590687.2	ADE34443.1	Turbot reddish body iridovirus
RBIV		AAT71848.1			Rock Bream iridovirus
**Ranavirus**					
FV3	NC_005946.1	ASH99239.1		AHM26101.1	Frog virus 3
EHNV	ACO25234.1	YP_009182042.1	AY187045.1	YP_009182084.1	Epizootic haematopoietic necrosis virus
GIV	AY666015.1		KX284838.1		Grouper iridovirus
ECV		AMZ05024.1		YP_006347705.1	European catfish virus
ATV		ALN36639.1		YP_003852.1	Ambystoma tigrinum virus
LMRV			KM516719.1		Lacerta monticola ranavirus
LMBV			KU507317.1		Largemouth bass ranavirus
**Lymphocystivirus**					
LCDV(Sa)		YP_009342128.1	AY823414.1		Lymphocystis disease virus (various strains)
LCDV(China)		YP_073534.1	NC_005902.1	YP_073585.1	
LCDV1				AAX54510.1	
**Iridovirus**					
IIV6	NC_003038.1	AAK82288.1		NP_149647.1	Invertebrate iridescent virus 6
IIV			NC_023615.1		Invertebrate iridescent virus
AVIV			NC_024451.1	NC_024451.1	Armadillium vulgare iridescent virus
Aedes taen.			NC_008187.1		Aedes taeniorhynchus iridescent virus
**Chloriridovirus**					
IIV3	YP_654692.1	YP_654581.1		YP_654693.1	Invertebrate iridescent virus 3
RMIV	CAC84133.1				Regular mosquito iridescent virus

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
