# Peer review of "Distribution and Phylogeny of Erythrocytic Necrosis Virus (ENV) in Salmon Suggests Marine Origin"

_viruses, 2019, doi:10.3390/v11040358_

Reviewer 1 Report

This is a well written manuscript. There are no major scientific flaws. Experiments are designed properly.

Minor:

the word "solely" in line 193 does sound funny. Hope authors will either rewrite the sentence or find an alternative word.

Figure 3 and 4 legends .. (a) and (b) should be changed to (A) and (B).

Author Response

Thank you for your recommendations, we have implemented the following changes:

the word "solely" in line 193 does sound funny. Hope authors will either rewrite the sentence or find an alternative word.

We have replaced the word “solely” with the word “only” in this line (now L 232). 

Figure 3 and 4 legends .. (a) and (b) should be changed to (A) and (B).

We have changed the capitalization in figures 3 and 4 accordingly. 

Reviewer 2 Report

Pagowaski et al have presented a manuscript provides details of erythrocytic necrosis virus prevalence in North Western America. This is valuable data specific to ENV that arises from a previous large surveillance study. The authors also generate novel virus sequence information that is used to infer some epidemiological features of this disease. The paper can be improved with editing to improve the description of the study, particularly to make the methods more readable and accessible. The results can be presented more clearly and a panel of figures to show the phylogeny. Care is needed to use terminology more precisely. Overall, the discussion does not have a sufficient number of references to link to the existing literature (e.g. L261 to 286 is a long and speculative section of text). A conclusion that links to the objective is required.

Specific comments:

Abstract: Quantities and values would be appropriate to support the qualitative statements about prevalence and genetic similarity. Include p-values for statistical comparisons where appropriate.

L38 and 51: Include scientific names

L52-54: A more circumspect statement without a reference to incidence and severity of disease impacting annual populations.

L57: resilient to disease/resistant to infection – distinguish

L60: The influence of host and environment factors is common to all diseases, it could be more clearly indicated that the sentence refers to pathogen genera-specific influences on disease manifestation.

L63: The definition of prevalence in parentheses is self-explanatory, perhaps the method used on the references should be the detail included to describe how prevalence was determined.

L86: Interrogating is an anthropomorphic term, reword.

L79: Further description of the survey design including time and location is necessary.

L88: More details are needed of the primers used to detect ENV in the samples and the sensitivity and specificity of the assay.

L98: Describe the selection process for samples that were sequenced.

L118: The sureselect enrichment process is not fully described. Perhaps bait libraries were informed by work performed in a stepwise approach which could be described more clearly?

Table 1: Sequence ID is not informative – this column might be replaced with some descriptive information about the source sample.

Figure 1: Include the length of the nucleotide fragment analysed in the caption.

Figure 2: Use the tem apparent prevalence.

Figure 3a: It would be useful to indicate the sampling effort in the survey area.

L235: This section of text would be more meaningful if the values of prevalence were discussed to indicate the magnitude of seasonal and long term changes.

L236: Indicate if this statement adjusted for age class/size in each season.

L249-250: Redundant sentence

L251: Strain variation was not determined – nucleotide/genotype variation was studies.

L255: replace of preface the term transcripts

L257: Without further explanation, this might be due to sequence variation or some other deficiency of the Sureselect enrichment procedure.

L263: The term incidence is not used appropriately; this would refer to new disease events occurring in given time periods not surveillance for infection with the pathogen.

L268: The data supporting the difference in aquaculture vs. wild prevalence is not clear in the results.

L326: add the worded ‘caused by’ to improve meaning.

L335: The virus would not change as different temperature – reword.

There is no clear conclusion linked to the objective.

The reason for the acknowledgment should be stated.

Author Response

Thank you for your comments that helped us improve the paper. We have amended the manuscript to address all of your concerns. A summary of your recommendations and our implemented changes is listed below.

Pagowaski et al have presented a manuscript provides details of erythrocytic necrosis virus prevalence in North Western America. This is valuable data specific to ENV that arises from a previous large surveillance study. The authors also generate novel virus sequence information that is used to infer some epidemiological features of this disease. The paper can be improved with editing to improve the description of the study, particularly to make the methods more readable and accessible. The results can be presented more clearly and a panel of figures to show the phylogeny. Care is needed to use terminology more precisely. Overall, the discussion does not have a sufficient number of references to link to the existing literature (e.g. L261 to 286 is a long and speculative section of text). A conclusion that links to the objective is required.

· We added a panel of phylogenies (Figure 1) and an additional table with details on the reference sequences and species used to create this figure. We also updated the supplementary figures to improve clarity. Some images are now in higher resolution and we have added a map of sampling locations (Figure S1). We have received additional sequencing data since submitting the original manuscript. Thus, file S4 has been updated accordingly.

· We have corrected terminology to clarify meaning, where applicable. The term “incidence” has been replaced with the word “prevalence” to maintain consistency. We have added the scientific names of species that we refer to in the manuscript (L38, 54-55). We have also re-worded L412-413 and 421-423 to be more specific in referring to virally-mediated disease progression.

· Additional references and description have been added to the discussion. Specifically, we have added references to support our claims in L334-337, 338-341, and 354-356.

· We added to the methods section by providing more detail about fish sampling (L92-95), data collection (L109-113), and the enrichment process (L140-149).

Abstract: Quantities and values would be appropriate to support the qualitative statements about

prevalence and genetic similarity. Include p-values for statistical comparisons where appropriate.

We have included quantitative statements to support our previous statements for the following claims:

· High genetic similarity between ENV isolated from salmon and sequences previously isolated from herring (nucleotide identity added, L24)

· High ENV prevalence in different species (percent prevalence in various marine species after LOD criteria applied added, L28-32)

· ENV prevalence is low in freshwater (p value from generalized linear mixed effects model test added, L30)

L38 and 51: Include scientific names

Scientific names are now included in L54-55

L52-54: A more circumspect statement without a reference to incidence and severity of disease impacting annual populations.

This statement has been rephrased, qualified, and citations added (all references updated accordingly). It now reads High geographic variability in VEN prevalence and disease susceptibility of chum, coho, sockeye, and Chinook salmon, as well as Pacific herring suggest that ENV could help explain high year-to-year variability in the population dynamics of these keystone species throughout coastal regions of the NE Pacific Ocean [1,7,8] (L56-59).

L57: resilient to disease/resistant to infection – distinguish

Rephrased as “resistant to infection”; this now reads (L62): Chinook, coho, and sockeye salmon appear to be more resistant to infection in challenge studies, as assessed by electron microscopy [5,11].

L60: The influence of host and environment factors is common to all diseases, it could be more clearly indicated that the sentence refers to pathogen genera-specific influences on disease manifestation.

This has been rephrased to (L66-67): There is substantial evidence that viruses in different genera the within the family Iridoviridae cause different disease manifestation and severity[10–15]

L63: The definition of prevalence in parentheses is self-explanatory, perhaps the method used on the references should be the detail included to describe how prevalence was determined.

Parentheses deleted, and we have rephrased a sentence in the methods to read (L115-117): ENV prevalence is reported as the proportion of fish with ENV detections, both with and without the LOD criteria applied. When not mentioned explicitly, prevalence values are reported with LOD criteria applied.

L86: Interrogating is an anthropomorphic term, reword.

L92: Replaced with “examining”

L79: Further description of the survey design including time and location is necessary.

The following has been added (L 93-95): Samples were collected over the course of an 11-year period from 2007-2018 in a region spanning Alaska to Northern Washington (Figure S1) as part of a large pathogen-screening effort conducted by Fisheries and Oceans Canada. The full dataset obtained by the DFO is not publicly available. However, we have included a supplementary figure (Figure S1) to indicate the overall sampling effort.

L88: More details are needed of the primers used to detect ENV in the samples and the sensitivity and specificity of the assay.

The following has been added (L 109-113): ENV assays on this platform show 100% inclusivity (detection of all known strain variants of the targeted microbe) and 97.9% exclusivity (no detection of untargeted microbe species). Primers were originally obtained from a 100-bp ATPase-like protein partial gene sequence. Additional details on primer sequence as well as assay specificity and reliability are available in Miller et al. [26].

L98: Describe the selection process for samples that were sequenced

The following has been added to clarify the selection process (L120-124): In order to isolate putative ENV sequences, several samples with high-load detections of this virus, as assessed using the Fluidigm BioMark, were selected for transcriptomic sequencing. In total, three aquaculture and one wild Chinook salmon were used to predict ENV proteins. Three additional fish, including an Atlantic salmon, a Chinook salmon, and a herring sample underwent an enrichment step for these predicted ENV proteins before bioinformatic analysis.

L118: The sureselect enrichment process is not fully described. Perhaps bait libraries were informed by work performed in a stepwise approach which could be described more clearly?

We have provided additional information (L141-149), so it is clear how the SureSelect approach that we used helped us validate putative ENV amino-acid sequences.

Table 1: Sequence ID is not informative – this column might be replaced with some descriptive information about the source sample.

The source sample has been added. The following has been added in the “Genetic Characterization” section (L227-232): Sequences reported in Table 1 originate from heart tissue of one of the aquaculture Chinook salmon samples and the wild Chinook salmon mixed tissue specimen, both with a high load ENV detection (CT values of 10.7 and 13.4, respectively). The aquaculture fish was observed to have jaundice and several coinfections including Paranucleospora theridion, Piscine reovirus, and Renibacterium salmoninarum, as determined by the RT-qPCR assay.

Figure 1: Include the length of the nucleotide fragment analysed in the caption.

The length of the nucleotide fragments used is included in Table 1. We have bolded these sequences and added a note in Figure 1 caption to clarify this.

Figure 2: Use the tem apparent prevalence.

“Proportion positive” has been replaced with “apparent prevalence” for all figures

Figure 3a: It would be useful to indicate the sampling effort in the survey area.

This has been indicated in a new supplementary figure (S1).

L235: This section of text would be more meaningful if the values of prevalence were discussed to indicate the magnitude of seasonal and long-term changes.

Quantitative statements were added to support claims of a large drop in prevalence after 2013, and higher prevalence in aquaculture fish. We have added the following (L303-304): A 63% decrease from the average prevalence of 5% was observed after 2013.

L236: Indicate if this statement adjusted for age class/size in each season.

This statement remains true if whether or not adults are removed from the sample. A spearman correlation was conducted with smolts and adults included for Atlantic salmon only as this species remains stationary throughout the year and adults represented a significant portion of the sample size. This is indicated in figure S3. We have added a statement specifying that this applies to both age classes.

L249-250: Redundant sentence

This sentence has been deleted (L309)

L251: Strain variation was not determined – nucleotide/genotype variation was studies.

Strain variation replaced with “nucleotide variation” (L309)

L255: replace of preface the term transcripts

This phrase has been deleted (L314)

L257: Without further explanation, this might be due to sequence variation or some other deficiency of the Sureselect enrichment procedure.

We have added the following (327-328): Moreover, our estimates of prevalence are conservative, as the assay is sensitive to sequence variation, so related strains of ENV could be missed.

L263: The term incidence is not used appropriately; this would refer to new disease events occurring in given time periods not surveillance for infection with the pathogen.

This has been replaced with the word “prevalence” (L337)

L268: The data supporting the difference in aquaculture vs. wild prevalence is not clear in the results.

This claim is supported by supplementary figure S4. We have added an explicit reference to this figure (L337)

L326: add the worded ‘caused by’ to improve meaning.

Sentence rephrased to the following (L 412-413): Numerous studies [10–14] have reported greater disease severity caused by iridoviruses that are closely related to ENV when temperature increases.

L335: The virus would not change as different temperature – reword.

This line has been reworded to suggest that the disease progression, rather than the virus itself, may change at different temperature (L421-423): It is possible that disease progression intensifies at warmer temperatures, such that fewer fish harboring the virus survive. This interpretation is consistent with a decrease in overall prevalence of salmon infectious agents in the region from 2012 to 2013 reported by Nekouei et al. [16].

There is no clear conclusion linked to the objective.

A conclusion has been added.

The reason for the acknowledgment should be stated.

An acknowledgment statement has been added.

Round  2

Reviewer 2 Report

The authors have responded to all issues raised in the review process.

Attention to detail is required fro typographical and grammatical errors  e.g. L62 delete "the"; Figure S6 Delete "Proportion" from y-axis label; L134 clarify "(497.266-kbp)".